# Abstraction Heuristics for Factored Tasks

**Primary Keywords:** *None*

## Abstract

One of the strongest approaches for optimal classical planning is A$^*$ search with heuristics based on abstractions of the planning task. Abstraction heuristics are well studied in planning formalisms without conditional effects such as SAS$^+$. However, conditional effects are crucial to model many planning tasks compactly. In this paper, we focus on *factored* tasks which allow a specific form of conditional effect, where effects on variable $x$ can only depend on the value of $x$. We generalize projections, domain abstractions, Cartesian abstractions and the counterexample-guided abstraction refinement method to this formalism. While merge-and-shrink already covers factored task in theory, we provide an implementation that does so. In our experiments, we compare these abstraction-based heuristics to other heuristics supporting conditional effects, as well as symbolic search. On our new benchmark set of factored tasks, pattern database heuristics solve the most problems, followed by symbolic approaches on par with domain abstractions. The more general Cartesian and merge-and-shrink abstractions fall behind.

## Introduction

In *classical planning*, we aim to find (short) paths in large, deterministic transition systems. In general, this means that we search for a sequence of actions leading from the initial state of the problem to a state which satisfies some goal condition. Interesting classical planning problems have too many states to find solutions using blind search. *Heuristic search* has proven a very successful method for solving classical planning problems (e.g., Bonet and Geffner 2001; Hoffmann and Nebel 2001; Helmert and Domshlak 2009; Richter and Westphal 2010; Helmert et al. 2014; Domshlak, Hoffmann, and Katz 2015). A *heuristic* is a function that estimates the cost from a given state to the closest goal state. The guidance of a good heuristic helps the search to focus on states that are likely part of a shortest solution.

One family of heuristics are *abstraction heuristics* (Seipp and Helmert 2018). An abstraction is an equivalence relation between states. The states of the original problem within the same equivalence class are mapped to a single abstract state. Each transition in the original problem induces a transition between the corresponding abstract states in the abstraction. Since the abstract problem generally has fewer states but preserves transitions between concrete states, it is easier to solve than the original problem. Moreover, the cost of a shortest path between two concrete states is lower-bounded by the cost of a shortest path between the corresponding abstract states. Abstraction heuristics can therefore use the abstract goal distance as an admissible estimate of the goal distance in the original problem. The most common abstraction classes in planning are *pattern database* (PDB) heuristics (Culberson and Schaeffer 1998), *domain abstractions* (Hernádvölgyi and Holte 2000), *merge-and-shrink (M&S) abstractions* (Dräger, Finkbeiner, and Podelski 2006), and *Cartesian abstractions* (Seipp and Helmert 2013). These methods differ in the way they map states to abstract states and can be ordered by increasing generality: PDBs, domain abstractions, Cartesian abstractions, and M&S abstractions (Seipp and Helmert 2018).

Compact representations of planning problems often require *conditional effects*. Unfortunately, many common heuristics including abstraction heuristics do not support conditional effects. We aim to fill this gap and study *factored tasks*, a class of planning tasks that augments SAS$^+$ tasks with limited forms of disjunctive preconditions, conditional effects, and angelic nondeterminism. While the theory on M&S already covers factored tasks, we show how the other abstraction classes mentioned above can be extended to this formalism. Furthermore, we implement these abstraction heuristics for the subset of factored tasks that can be modeled in finite-domain representation (Helmert 2009). We also provide a set of benchmark problems consisting of the well-known permutation puzzles *Pancakes*, *Burnt Pancakes*, *Rubik's Cube*, and *TopSpin* as well as the recently introduced domain for finding algorithms for *Matrix Multiplication* (Speck et al. 2023).

In our experiments, we compare our abstraction heuristics with previous approaches for solving planning problems with conditional effects, such as $h^{\mathrm{max}}$ (Bonet and Geffner 2001), LM-cut with context splitting (Röger, Pommerening, and Helmert 2014) and symbolic search (Edelkamp, Kissmann, and Torralba 2015; Torralba, Linares López, and Borrajo 2016). The most problems are solved with PDBs, followed by domain abstractions on par with symbolic approaches. The more general Cartesian and M&S abstractions fall behind in terms of coverage. Most likely, this is because computing meaningful abstractions for our benchmarks requires more resources than we provide, rendering the simpler PDB and domain abstractions preferable.

## Factored Tasks

We consider classical planning with a factored task representation (Helmert et al. 2014; Torralba and Sievers 2019; Sievers and Helmert 2021). Earlier papers introduce factored tasks with an automata-based representation. Here we give an equivalent definition that is closer in spirit to planning task representations like STRIPS and $SAS^+$.

Factored tasks extend the $SAS^+$ formalism (Bäckström and Nebel 1995) with conditional effects, disjunctive preconditions and goal conditions, and angelic nondeterminism. However, all three features are restricted in such a way that the structure of tasks can be understood by considering *one variable at a time*: for example, each conditional effect on a variable only depends on this variable.

A *variable space* is a tuple $\mathcal{V} = \langle V_1, \ldots, V_n \rangle$ of variables with a finite domain. We write $dom(V_i)$ for the domain of $V_i$, which can be an arbitrary finite set of values. A *state* of $\mathcal{V}$ is a tuple $\langle d_1, \ldots, d_n \rangle$ with $d_i \in dom(V_i)$ for all $1 \leq i \leq n$. It follows that the set of all states is the Cartesian product $dom(V_1) \times \cdots \times dom(V_n)$. We denote this set by $[\![\mathcal{V}]\!]$.

A *factored state set* for $\mathcal{V}$ is a tuple $D = \langle D_1, \ldots, D_n \rangle$ of subsets $D_i \subseteq dom(V_i)$. It serves as a compact representation of the (non-factored) state set $[\![D]\!] = D_1 \times \cdots \times D_n \subseteq [\![\mathcal{V}]\!]$. State sets represented by factored state sets are called *Cartesian sets* (Seipp and Helmert 2018).

A *factored state relation* for $\mathcal{V}$ is a tuple $R = \langle R_1, \ldots, R_n \rangle$ of *relations* $R_i \subseteq dom(V_i) \times dom(V_i)$. It serves as a compact representation of the (non-factored) relation $[\![R]\!] \subseteq [\![\mathcal{V}]\!] \times [\![\mathcal{V}]\!]$ with $\langle s, s' \rangle \in [\![R]\!]$ iff $\langle s_i, s_i' \rangle \in R_i$ for all $1 \leq i \leq n$. Factored state relations are a natural generalization of factored state sets in the sense that we can determine membership in the relation by conducting separate tests for each variable. A *factored operator* $o$ is defined by a factored state relation $trans(o)$ and its operator cost $cost(o)$.

A *factored task* is a 4-tuple $\Pi = \langle \mathcal{V}, \mathcal{O}, I, G \rangle$, where $\mathcal{V}$ is a variable space, $\mathcal{O}$ is a finite set of factored operators, and $I$ and $G$ are factored state sets representing the initial and goal states.

The semantics of tasks are defined via *transition systems*. A transition system is a 5-tuple $\mathcal{T} = \langle \mathcal{S}, \mathcal{O}, T, S_\mathrm{I}, S_\mathrm{G} \rangle$, where $\mathcal{S}$ is a finite set of states, $\mathcal{O}$ is a finite set of operator labels[1] with associated cost, $T \subseteq \mathcal{S} \times \mathcal{O} \times \mathcal{S}$ is a finite set of labeled transitions, and $S_\mathrm{I}, S_\mathrm{G} \subseteq \mathcal{S}$ are the initial and goal states.

The factored task $\Pi = \langle \mathcal{V}, \mathcal{O}, I, G \rangle$ represents the transition system $[\![\Pi]\!] = \langle [\![\mathcal{V}]\!], \mathcal{O}, [\![\mathcal{O}]\!], [\![I]\!], [\![G]\!] \rangle$, where $[\![\mathcal{O}]\!] = \{ \langle s, o, s' \rangle \mid o \in \mathcal{O}, \langle s, s' \rangle \in [\![trans(o)]\!] \}$. The objective of classical planning is to find a path from an initial to a goal state in $[\![\Pi]\!]$. The multiple initial states and nondeterminism of operators (a single state can have multiple successors via the same operator) are interpreted angelically, i.e., the planning algorithm may choose which initial state/successors to use (Torralba and Sievers 2019).

---

[1] We call these *operator labels* instead of *operators* to emphasize that within a transition system, they only serve as opaque labels rather than objects with internal structure like the operators of a factored task.

Note that factored tasks may have an empty set of initial states or goal states. If so, they are trivially unsolvable because there cannot exist a path from an initial to a goal state in these cases. We say a factored task with $n$ variables is *trivially unsolvable* if $I_i = \emptyset$ or $G_i = \emptyset$ for some $1 \leq i \leq n$. A *trivial operator* is an operator with $R_i = \emptyset$ for some $1 \leq i \leq n$ which therefore induces no transitions in the represented transition system (Sievers and Helmert 2021).

We remark that we need neither multiple initial states nor (angelically) nondeterministic operators for this paper, but they can be supported at no additional difficulty and allow us to treat some things more generally and more uniformly. In particular, they make *regression* very simple for factored tasks: by swapping the initial and goal states and replacing each transition relation $trans(o)_i$ by its inverse $trans(o)_i^{-1}$, we obtain a new factored task whose transition system is the inverse of the original transition system.

**Comparison to $SAS^+$** Factored tasks generalize $SAS^+$ tasks. $SAS^+$ tasks can be understood as factored tasks $\langle \langle V_1, \ldots, V_n \rangle, \mathcal{O}, I, G \rangle$ with the following restrictions:

- There is a single initial state: $|I_i| = 1$ for all $1 \leq i \leq n$.
- Variables either have a single goal value or no goal condition: $|G_i| = 1$ or $G_i = dom(V_i)$ for all $1 \leq i \leq n$.
- For all $o \in \mathcal{O}$ and all $1 \leq i \leq n$, the relation $R_i = trans(o)_i$ has one of the following forms:
  - $R_i = \{ \langle d, d \rangle \mid d \in dom(V_i) \}$: the operator has no precondition or effect on $V_i$
  - $R_i = \{ \langle d, d \rangle \}$ for exactly one $d \in dom(V_i)$: the operator has a precondition and no effect on $V_i$
  - $R_i = \{ \langle d, d' \rangle \}$ for exactly one pair $d, d' \in dom(V_i)$: the operator has a precondition and an effect on $V_i$
  - $R_i = \{ \langle d, d' \rangle \mid d \in dom(V_i) \}$ for exactly one $d' \in dom(V_i)$: the operator has an effect and no precondition on $V_i$

For tasks that can be compactly expressed in $SAS^+$, factored task representations are somewhat more verbose for aspects such as variables not appearing in a precondition or effect. This is not a concern for this paper, but we note that practical implementations sometimes special-case these aspects to reduce verbosity (Sievers 2018).

Conversely, factored tasks allow representing some aspects compactly that cannot be directly represented in $SAS^+$. For example, if $dom(V_i) = \{1, 2, 3, 4, 5\}$ an operator can use the relation $R_i = \{ \langle 2, 3 \rangle, \langle 3, 2 \rangle, \langle 4, 4 \rangle \}$ to express the disjunctive precondition $(V_i = 2) \vee (V_i = 3) \vee (V_i = 4)$ and the conditional effects $(V_i = 2) \rhd (V_i := 3)$ and $(V_i = 3) \rhd (V_i := 2)$. Expressing the same transition semantics in $SAS^+$ requires either exponential-size compilation or introducing auxiliary state variables and operators split into multiple stages, which can negatively affect planning algorithms (Nebel 2000).

The same is true for the angelic nondeterminism supported by factored tasks. For example, we can easily express an operator with the meaning "For each of the variables $V_1, \ldots, V_k$, choose either 1 or 2 as the new value", but this would require $2^k$ operators or an operator split into multiple stages using auxiliary states in $SAS^+$.

## Abstractions

Most current planning algorithms use heuristic search algorithms to perform a progression search through the transition system $[\![\Pi]\!]$. Two such heuristic search algorithms are A$^*$ (Hart, Nilsson, and Raphael 1968) and IDA$^*$ (Korf 1985), which use a *heuristic function* to estimate the cost to reach the goal from each search node. They guarantee an optimal (minimum-cost) solution if the heuristic is *admissible*, i.e., never overestimates the cost to the goal.

*Abstractions* are a common source of admissible heuristics in the planning literature. The four most widely studied classes of abstractions are *projections* used for pattern database (PDB) heuristics (Edelkamp 2001), *domain abstractions* (Hernádvölgyi and Holte 2000), *Cartesian abstractions* (Seipp and Helmert 2018), and *merge-and-shrink (*M&S*) abstractions* (Sievers and Helmert 2021). In this section we introduce the general concept of abstraction and these specific classes. In the following section, we extend heuristics using these classes of abstractions from SAS$^+$ to factored tasks.

Let $\mathcal{T} = \langle \mathcal{S}, \mathcal{O}, T, S_{\mathrm{I}}, S_{\mathrm{G}} \rangle$ be a transition system. An *abstraction* $\sim$ is an equivalence relation over $\mathcal{S}$ with the meaning that the distinction between states in the same equivalence class is ignored. We write $s^\sim$ for the equivalence class to which state $s \in \mathcal{S}$ belongs and define $S^\sim = \{s^\sim \mid s \in S\}$ for sets of states $S$.

The abstraction $\sim$ induces the *abstract transition system* $\mathcal{T}^\sim = \langle \mathcal{S}^\sim, \mathcal{O}, T^\sim, S_{\mathrm{I}}^\sim, S_{\mathrm{G}}^{\widetilde{~}} \rangle$ where $T^\sim = \{\langle s^\sim, o, t^\sim \rangle \mid \langle s, o, t \rangle \in T\}$. The *abstraction heuristic* $h^\sim$ maps state $s$ to the minimum path cost from $s^\sim$ to any $t^\sim \in S_{\mathrm{G}}^{\widetilde{~}}$ in $\mathcal{T}^\sim$. By construction, every path in $\mathcal{T}$ corresponds to a path in $\mathcal{T}^\sim$. Consequently, the minimum path cost from $s^\sim$ to $t^\sim$ in $\mathcal{T}^\sim$ is a lower bound on the minimum path cost from $s$ to $t$ in $\mathcal{T}$. Together with the definition of the abstract goal states, it follows that abstraction heuristics are admissible.

Abstraction heuristics for planning exploit that the set of states $S$ is represented by a variable space, i.e., $S = [\![\mathcal{V}]\!]$ for some $\mathcal{V} = \langle V_1, \ldots, V_n \rangle$.

**Pattern Databases**   PDBs are based on projections onto a subset of the variables $P \subseteq \{V_1, \ldots, V_n\}$ called the *pattern*. Two states $s$ and $t$ are equivalent in the abstraction iff $s_i = t_i$ for all $V_i \in P$. Sievers, Ortlieb, and Helmert (2012) explain how to compute PDBs efficiently for SAS$^+$ tasks and how to support efficient heuristic computation via table lookup.

**Domain Abstractions**   In a projection, each state variable is either represented faithfully (for variables in the pattern) or not at all (for variables outside the pattern). *Domain abstractions* generalize this idea by defining an equivalence relation $\sim_i \subseteq dom(V_i) \times dom(V_i)$ for each state variable $V_i$. Two states $s$ and $t$ are equivalent under such a domain abstraction if they are equivalent in each of these relations: $s \sim t$ if $s_i \sim t_i$ for all state variables $V_i$.

Projections can be expressed as domain abstractions by using the identity relation $\{\langle d, d \rangle \mid d \in dom(V_i)\}$ as the equivalence relation for variables in the pattern (all values of the variable are distinguished) and the universal relation $dom(V_i) \times dom(V_i)$ for variables outside the pattern (all values are considered equivalent).

Kreft et al. (2023) describe a state-of-the-art implementation of domain abstraction heuristics for SAS$^+$ tasks based on the counterexample-guided abstraction refinement principle (CEGAR).

**Cartesian Abstractions**   An abstraction $\sim$ is called *Cartesian* if all equivalence classes under $\sim$ are Cartesian sets. Domain abstractions (and therefore also projections) are a special case of Cartesian abstractions: if we consider a domain abstraction with equivalence relations $\sim_i$ for the individual variables, then all equivalence classes are of the form $D_1 \times \cdots \times D_n$, where $D_i$ is an equivalence class of $\sim_i$.

Cartesian abstractions strictly generalize domain abstractions because they do not require a *global* decision on how to partition variable domains into equivalence classes. The decision which values of state variables are grouped together is made individually at the level of each abstract state.

Seipp and Helmert (2018) describe an efficient implementation of Cartesian abstractions for SAS$^+$ tasks based on the CEGAR principle.

**Merge-and-Shrink Abstractions**   The most general class of abstractions we consider are *merge-and-shrink (*M&S*) abstractions* (Sievers and Helmert 2021). To generate an M&S abstraction, we begin with a pool of abstract transition systems that represent all projections to a single state variable. This pool of transition systems is repeatedly transformed by replacing two transition systems with their product (*merging*) and reducing the size of a transition system by applying a local abstraction (*shrinking*) until only a single abstract transition system remains, which then defines the abstraction heuristic.[2]

M&S can represent arbitrary abstractions, which makes this approach even more general than Cartesian abstractions. However, not all abstractions can be represented compactly in the merge-and-shrink framework, and in this regard the precise relationship to Cartesian abstractions is an open question. M&S abstractions are known to properly generalize pattern database heuristics for SAS$^+$ tasks, also in the sense that the computation is similarly compact and efficient (Helmert, Haslum, and Hoffmann 2007; Sievers and Helmert 2021), and it is easy to extend this result to domain abstractions. The existing theory of M&S abstractions covers the full generality of factored tasks (Helmert et al. 2014; Sievers and Helmert 2021), but the implementations described in the literature are limited to SAS$^+$.

## Abstraction Heuristics for Factored Tasks

We now describe how the four classes of abstractions can be extended from SAS$^+$ tasks to factored tasks. The main contributions are the extensions for domain abstractions and counterexample-guided Cartesian abstraction refinement (Cartesian CEGAR) because the PDB case is straightforward and the M&S case is already covered in the literature.

---

[2]The full algorithm also applies further transformations called *label reduction* and *pruning*, which are not important for this discussion (Sievers and Helmert 2021).

**Pattern Databases**  The efficient implementation of pattern databases for SAS$^+$ tasks $\Pi$ is based on the idea of *syntactic projection*: remove all state variables that are not part of the pattern from the compact task representation, then use the resulting task to produce the abstract transition system. For SAS$^+$ tasks, this approach is *conservative* (every transition of $[\![\Pi]\!]$ has a corresponding transition in the abstract transition system) and *induced* (every transition in the abstract transition system has a corresponding transition in $[\![\Pi]\!]$) and hence results in exactly the transition system $[\![\Pi]\!]^\sim$.

For more general classes of planning tasks, this is not necessarily the case. Consider the pattern $\{V_2, V_3\}$ and an operator with the unconditional effect $(V_2 := 1)$ and the conditional effect $(V_1 = 0) \rhd (V_3 := 1)$. If we naively define the syntactic projection to receive the unconditional effects $(V_2 := 1)$ and $(V_3 := 1)$, the resulting abstract transition system misses transitions that should be present: in $\Pi$ the operator can sometimes change $V_2$ without changing $V_3$, but in the projected task it cannot. There are ways to avoid this problem, but they all have a price such as accepting non-induced abstractions (reducing heuristic quality) or making it NP-hard to test if an induced transition exists.

For factored tasks, this problem does not arise because dependencies between different state variables as in the problematic conditional effect do not exist. For factored tasks, syntactic projection is conservative and leads to an induced abstraction with one caveat: if the problem is trivially unsolvable or has trivial operators, projecting away the responsible variables may lead to solvable abstractions or non-induced transitions of the operator. Hence, we need to check that the sets of original initial and goal states are nonempty and discard all operators $o$, for which $[\![trans(o)]\!] = \emptyset$. This is easy to do in linear time in the size of the task representation. We do not prove this result formally because it follows from the general relationship between PDBs and M&S (Sievers and Helmert 2021).

**Domain Abstractions**  Like PDBs, domain abstractions can be implemented for SAS$^+$ as *syntactic domain abstractions*. We assign a value between 1 and the number of equivalence classes to each equivalence class in $\sim_i$. Then, we replace all variable values with the number representing their corresponding equivalence class wherever they occur in the task representation. Again, this approach is conservative and leads to an induced abstraction, resulting exactly in $[\![\Pi]\!]^\sim$. Extending it to general conditional effects comes with the same problems as PDBs.

For domain abstractions of factored tasks, we can again point out the relationship to M&S and use the more general result to show that they are conservative and induced. Starting with the set of transition systems of atomic projections, we can first shrink all these factors according to the equivalence relations in each variable domain. From Sievers and Helmert (2021) we know that shrinking is an induced and conservative transformation. Now we can merge the factors same as in the PDB case, to end up with a transition system that is induced and conservative. Moreover, it is isomorphic to the transition system obtained through syntactic domain abstraction. Note that the caveat we mention for PDBs does

not apply here: If a factored state set or relation is empty in the original problem, then it is also empty in the syntactic domain abstraction where variable values are simply replaced by their abstract values.

Another notable difference to the case of PDBs is the following observation: abstract transition systems may be nondeterministic even if the original transition system is deterministic. Consider for example a factored task with a single variable $V_1$ with $dom(V_1) = \{x, y, z\}$ and an operator $o = \langle \{\langle x, y \rangle, \langle y, z \rangle \} \rangle$. There is at most one outgoing transition labeled with $o$ in every state of the induced transition system. Now let $x \sim_1 y$ and $x \not\sim_1 z$ in the domain abstraction and let 1 denote the equivalence class of $x$ and $y$ and 2 denote the equivalence class of $z$. We end up with the relation $R_1^\sim = \{\langle 1, 1 \rangle, \langle 1, 2 \rangle\}$ which means there are two outgoing transitions labeled with $o$ in the abstract state representing $V_1 = 1$. This observation is one of the justifications for considering angelic nondeterminism in our factored task representation.

**Cartesian Abstractions**  In the literature, Cartesian abstractions are usually discussed together with the algorithm used to compute them: Cartesian CEGAR (Seipp and Helmert 2018). Cartesian CEGAR imposes a hierarchical structure on the Cartesian sets representing the abstract states. This structure plays an important role regarding the efficiency of the algorithm and the heuristic lookup after its termination. It remains an open question whether and how we can deal with Cartesian abstractions that does not follow such a hierarchy.

Because Cartesian abstractions generalize domain abstractions and projections, extending the Cartesian CEGAR algorithm to more expressive classes of planning tasks leads to similar problems and other problems besides. However, the algorithm can be efficiently extended to factored tasks.

Consider the transition system $\mathcal{T} = \langle \mathcal{S}, \mathcal{O}, T, S_I, S_G \rangle$, an operator label $o \in \mathcal{O}$ and a state set $S \subseteq \mathcal{S}$. We define:

- the *progression* of $S$ through operator $o$:
  $progr(S, o) = \{s' \mid s \in S, \langle s, o, s' \rangle \in T\}$
- the *regression* of $S$ through operator $o$:
  $regr(S, o) = \{s \mid s' \in S, \langle s, o, s' \rangle \in T\}$
- the set of states in which $o$ is *applicable*:
  $applicable(o) = \{s \mid \langle s, o, s' \rangle \in T\}$

Our main observation is that the Cartesian CEGAR algorithm can be extended to any transition system with the following properties:

(P1)  $S_I$ and $S_G$ are Cartesian sets.

(P2)  For every Cartesian set $S$ and every operator label $o$, $progr(S, o)$ and $regr(S, o)$ are Cartesian sets.

(P3)  For every $o \in \mathcal{O}$, $applicable(o)$ is a Cartesian set.

Moreover, the algorithm can be made as efficient as in the SAS$^+$ case if all Cartesian sets are represented as factored state sets and $progr(S, o)$ and $regr(S, o)$ have efficient implementations for such representations.

Factored tasks satisfy all these properties. Let $\Pi = \langle \langle V_1, \ldots, V_n \rangle, \mathcal{O}, I, G \rangle$ be a factored task. Property (P1) is obvious because $I$ and $G$ are factored state sets.

**Algorithm 1:** Abstract trace verification. Try to convert the given abstract solution into a solution for the planning task. If this fails, return a flaw of the form $\langle A, B_1, B_2 \rangle$: a Cartesian set $A$ that must be split to separate $B_1 \subseteq A$ from $B_2 \subseteq A$.

---

**1 function** FINDFLAW($\langle A_0, o_1, A_1, \ldots, o_n, A_n \rangle$)
**2**    $Poss \leftarrow A_0 \cap S_I$
**3**    **for** $i = 1$ **to** $n$ **do**
**4**        { *invariants:* $Poss \neq \emptyset$, $Poss \subseteq A_{i-1}$ }
**5**        **if** $Poss \cap applicable(o_i) = \emptyset$ **then**
**6**            { *flaw found:* violated precondition }
**7**            **return** $\langle A_{i-1}, Poss, A_{i-1} \cap applicable(o_i) \rangle$
**8**        **if** $progr(Poss, o_i) \cap A_i = \emptyset$ **then**
**9**            { *flaw found:* cannot get to next abstract state }
**10**           **return** $\langle A_{i-1}, Poss, A_{i-1} \cap regr(A_i, o_i) \rangle$
**11**        $Poss \leftarrow progr(Poss, o_i) \cap A_i$
**12**    **if** $Poss \cap S_G = \emptyset$ **then**
**13**        { *flaw found:* goal not reached }
**14**        **return** $\langle A_n, Poss, A_n \cap S_G \rangle$
**15**    **return** "no flaw"

---

For property (P2), consider $S = D_1 \times \cdots \times D_n$ and operator $o$ with $trans(o) = \langle R_1, \ldots, R_n \rangle$. Because factored tasks consider each state variable separately, we get

$$progr(S, o) = D_1' \times \cdots \times D_n', \text{ where}$$
$$D_i' = \{d' \in dom(V_i) \mid d \in D_i, \langle d, d' \rangle \in R_i\}.$$

Regression is analogous. Intuitively, Cartesian sets can be progressed/regressed through operators of factored tasks by progressing/regressing each state variable separately.

Property (P3) can be seen analogously, but also follows from (P2) in general because $applicable(o) = regr(\mathcal{S}, o)$ for all operators $o$, and the set of all states $\mathcal{S}$ is Cartesian.[3]

We now show how to extend Cartesian CEGAR using the three properties. As a reminder, Cartesian CEGAR is an iterative algorithm that maintains an abstract transition system $\mathcal{T}$ represented as an explicit digraph. Each iteration of the algorithm finds an optimal solution (minimum-cost path from an initial state to a goal state) for $\mathcal{T}$ and checks whether this solution works in the real planning task. If not, the algorithm determines a flaw (a reason *why* the solution does not work) and addresses it by splitting one abstract state into two abstract states.

Checking the solution and finding a flaw is the responsibility of the function FINDFLAW, which forms the heart of the Cartesian CEGAR approach. We focus our discussion on this function; all other components of the approach are straightforward to adapt. Algorithm 1 shows FIND-FLAW for the factored task setting. The original version for SAS$^+$ tasks is Algorithm 2 in the paper of Seipp and Helmert (2018).

For simplicity, the algorithm is written as if it operated directly on Cartesian sets. In the implementation, these are

---

[3]Of course this makes (P3) a redundant property. We find it useful to state nevertheless because we use it in the following.

---

represented as factored state sets, which efficiently support the necessary operations such as set intersection and comparison to the empty set. Note that the algorithm uses properties P1–P3, as it uses all of $S_I$, $S_G$, *progr*, *regr* and *applicable* and requires the sets it operates on to be Cartesian. Note also that apart from the properties P1–P3, the algorithm is completely agnostic to the task representation.

The input to the algorithm is the found abstract solution, represented as a *trace*, i.e., the abstract states (= Cartesian sets) $A_0, \ldots, A_n$ that form the path interleaved with the operator labels $o_1, \ldots, o_n$ that label the used transitions between these abstract states. The case $n = 0$ is allowed, in which case the trace is simply $\langle A_0 \rangle$, which must then be both an abstract initial state and an abstract goal state.

FINDFLAW verifies the trace step by step. At any point, $Poss$ is a Cartesian set representing the concrete states that the part of the trace that was verified so far can lead to. (In the original algorithm by Seipp and Helmert, this is always a single state because SAS$^+$ tasks have a single initial state and deterministic operators.) $Poss$ is always nonempty: if at any stage it would become empty, this signifies a flaw, and the algorithm returns.

There are three kinds of flaws that can arise: 1) violated preconditions, 2) the real solution leading to a different abstract state than the abstract trace does, and 3) not ending in a goal state. The conditions for detecting these flaws are very similar to the original algorithm, and we refer to Seipp and Helmert (2018) for a detailed discussion.

Whenever a flaw is found, the algorithm identifies three Cartesian sets $A$, $B_1$ and $B_2$. The set $A$ is an abstract state in the current abstraction that must be split to repair the flaw. The set $B_1 \subseteq A$ consists of the states of the planning task that the verified part of the trace leads to; the set $B_2 \subseteq A$ consists of the states that would have needed to be reached in order to continue with the verification of the trace.

The sets $B_1$ and $B_2$ are always nonempty and disjoint. To repair the flaw, Cartesian CEGAR selects a variable for which the values allowed in $B_1$ and $B_2$ are disjoint (such a variable must exist because $B_1$ and $B_2$ are disjoint) and uses it to partition $A$ into two new Cartesian sets $A_1$ and $A_2$ with $B_1 \subseteq A_1$ and $B_2 \subseteq A_2$. The algorithm removes $A$ from the abstract transition system and replaces it with $A_1$ and $A_2$, adding the necessary transitions and marking the new abstract states as initial and goal states as needed. This concludes the iteration of the CEGAR loop and our discussion of Cartesian CEGAR.

**Merge-and-Shrink** For M&S, the existing theory already covers factored tasks and therefore does not need to be extended (Helmert et al. 2014; Sievers and Helmert 2021). However, the existing *implementations* described in the literature require SAS$^+$. We extended the implementation to support the kinds of conditional effects supported in factored tasks, but this does not require new theory.

## Experiments

Given the success of abstraction-based planning algorithms for SAS$^+$ tasks, it is natural to ask whether these approaches also work well for factored tasks. To evaluate

this, we implement the necessary extensions in the Scorpion planner (Seipp, Keller, and Helmert 2020), an extension of Fast Downward (Helmert 2006). The task representation expected as input to these systems does not support multiple initial states, disjunctive preconditions and goal conditions, nor angelic nondeterminism. While it would be interesting to consider these generalizations as well, here we focus solely on factored conditional effects. By considering only this addition we can easily compare our algorithms to existing planners that support conditional effects but not necessarily the other features of factored tasks.

We use Downward Lab (Seipp et al. 2017) for running our experiments with a time limit of 30 minutes and a memory limit of 8 GiB. In the following, we describe the tasks we use for our evaluation, give an overview of the compared planner configurations, and evaluate their performance. Our code, benchmarks, and data will be published online (Reference omitted for anonymity).

## Benchmarks

For our analysis, we implemented problem generators that create finite-domain representations (Helmert 2009) of the following domains: *Pancakes* (e.g., Dweighter 1975) and its variation *Burnt Pancakes* (e.g., Gates and Papadimitriou 1979), *Rubik's Cube* (e.g., Korf 1997), and *TopSpin* (e.g., Holte et al. 2006). For each of them, we generate 100 problems of varying difficulty. Furthermore, we use the 11 benchmarks introduced by Speck et al. (2023) for finding algorithms for *Matrix Multiplication*.

Permutation problems such as the ones in the list above can be modeled naturally as factored tasks. Consider, for example, the *Pancakes* domain where a stack of pancakes must be ordered by size by inserting a spatula into the stack and flipping all pancakes above that point. There are $\frac{n!}{(n-k)!}$ possible situations for the top $k$ pancakes. Hence, modeled as a $SAS^+$ task with one variable for every pancake, a problem with $n$ pancakes requires $\sum_{k=1}^{n} \frac{n!}{(n-k)!}$ operators. In finite-domain representation, we only require $n$ operators, one for every $k$ with conditional effects for changing the position of every pancake depending on its current position.

Besides the possibility to model these kinds of problems compactly as factored tasks, the selected domains are interesting for our analysis because abstraction heuristics are considered the state of the art for solving some of these and similar problems (e.g., Korf 1997; Hernádvölgyi and Holte 2000; Korf and Felner 2002). While the state-of-the-art methods are domain-specific, it is interesting to see how our domain-independent approaches perform on these kinds of problems.

## Heuristics

We use the following heuristics.

**Blind Heuristic** Our baseline is blind search (BLIND).

$h^{\mathrm{max}}$ **Heuristic** A common admissible heuristic that supports conditional effects is the $h^{\mathrm{max}}$ heuristic (Bonet and Geffner 2001) which we denote by HMAX in our evaluation.

**Pattern Database Heuristics** For PDBs, the crucial decisions are how to choose the patterns and how to combine their individual heuristic values. For the generation of pattern collections we use the following two approaches:

PDB-SYS This configuration systematically generates all *interesting* patterns up to a certain size (Pommerening, Röger, and Helmert 2013). For our benchmarks, this strategy is feasible for patterns of size up to 3.

PDB-CEGAR Rovner, Sievers, and Helmert (2019) propose generating patterns using the CEGAR algorithm. We adapt it to work for factored tasks but use their recommended settings: The time limit for generating patterns is 100 seconds, each induced abstractions may contain at most 1M states while the entire collection is limited to 10M states, and blacklisting triggers after 75 seconds or if no new patterns are found for 20 seconds.

We combine the abstract goal distances of each pattern by taking the maximum. We also tested combining them with *saturated cost partitioning* (SCP; Seipp, Keller, and Helmert 2020), a state of the art approach for combining abstraction heuristics in $SAS^+$, but taking the maximum performed significantly better on our benchmarks, so we only report results for the maximum below.

**Domain Abstraction Heuristic** Using CEGAR was also suggested for generating domain abstractions (Kreft et al. 2023). Extending it to factored tasks is not as straightforward as for PDBs because, as we mention above, domain abstractions may become nondeterministic even if the original problem is deterministic (which is the case for our benchmarks). This introduces a type of flaw that cannot occur in the $SAS^+$ case or the PDB case, but we have handled it above for Cartesian CEGAR: Applying the abstract plan to the original problem may diverge from the abstract trace. Incorporating this in the algorithm is easy enough, by storing not only the abstract plan but also the sequence of abstract states and returning a flaw as soon as the concrete state does no longer correspond to the abstract state when executing the abstract plan in the concrete problem. We consider two configurations of domain abstractions:

DOM-SINGLE This strategy computes a single domain abstraction with at most 1M states using the recommended refinement strategy of picking flaws randomly.

DOM-MULTI In this configuration, we compute a collection of domain abstractions such that individual abstractions have at most 10K states while the entire collection may have up to 1M states. To get diverse abstractions, we initialize each run by choosing an arbitrary goal variable and represent it with the identity relation in the initial abstraction. The refinement strategy is again picking one random flaw in every iteration. Blacklisting is activated from the start. During search we maximize over all individual estimates.

**Cartesian Abstraction Heuristic** By CARTESIAN we denote the Cartesian CEGAR algorithm adapted to factored tasks in this paper. We limit neither the number of states nor transitions. The abstraction refinement loop terminates

| | BLIND | HMAX | LM-CUT | SYMBA* | SYMBB | Abstraction Heuristics | | | | | |
|---|---|---|---|---|---|---|---|---|---|---|---|
| | | | | | | PDB-SYS | PDB-CEGAR | DOM-SINGLE | DOM-MULTI | CARTESIAN | M&S |
| Pancakes (100) | 39 | 39 | 35 | 53 | 52 | 52 | **59** | 49 | 51 | 44 | 43 |
| Burnt Pancakes (100) | 35 | 38 | 30 | 49 | 49 | 49 | **53** | 44 | 47 | 40 | 40 |
| Rubik's Cube (100) | 36 | 42 | 35 | 56 | 50 | 59 | **66** | 51 | 58 | 47 | 46 |
| TopSpin (100) | 24 | 31 | 22 | 50 | 51 | 49 | **59** | 45 | 49 | 44 | 32 |
| Matrix Multiplication (11) | **7** | **7** | **7** | **7** | **7** | **7** | **7** | **7** | **7** | **7** | **7** |
| Total (411) | 141 | 157 | 129 | 215 | 209 | 216 | **243** | 196 | 212 | 182 | 168 |
| out of memory | 270 | 1 | – | 78 | 52 | 23 | 107 | 215 | 90 | 100 | 240 |
| out of time | – | 253 | 282 | 118 | 150 | 172 | 61 | – | 109 | 129 | 3 |

Table 1: Number of solved tasks per domain and summary of reasons for failure.

once the refinement finds a valid plan, 900 seconds passed, or less than 500 MiB of free memory remain. We also considered computing collections of Cartesian abstractions, one for each goal fact (Seipp and Helmert 2014), but both taking their maximum or computing an SCP performed significantly worse than just using a single Cartesian abstraction, so we do not report these numbers.

**Merge-and-Shrink Heuristic** The last abstraction-based configuration we consider is merge-and-shrink (M&S). It uses bisimulation as its shrink strategy (Nissim, Hoffmann, and Helmert 2011), strongly connected components as the merge strategy (Sievers, Wehrle, and Helmert 2016), and exact label reduction (Sievers and Helmert 2021). We limit the abstraction size to at most 50K states.

## Planners

We run an A* search for each of the heuristics above. In addition, we evaluate:

**A* with Context-Splitting LM-Cut** By LM-CUT we denote an A* search using the landmark-cut heuristic extended with context-splitting to support conditional effects (Röger, Pommerening, and Helmert 2014). We use the implementation from Metis (Sievers and Katz 2018) which also supports partial order reduction based on strong stubborn sets (Wehrle and Helmert 2014) and structural symmetry pruning based on orbit space search (Shleyfman et al. 2015; Domshlak, Katz, and Shleyfman 2015). Since our experiments showed that both of these pruning methods fail to prune the state spaces of our benchmarks, we only report results for plain LM-Cut with context splitting.

**Symbolic Search** Finally, we evaluate two planners based on symbolic search. First, we consider SYMBA*, winner of the optimal track of IPC 2014 (Torralba, Linares López, and Borrajo 2016). Second, we run symbolic bidirectional blind search (SYMBB) (Edelkamp, Kissmann, and Torralba 2015) using the implementation from SymK (Speck 2022).

## Coverage Evaluation

Table 1 shows how many problems of each domain are solved by the approaches above and the reasons for failure. In all domains, PDB-CEGAR solves the most problems. Summarizing coverage over all domains, PDB-CEGAR solves many more tasks (243 tasks) than its closest contenders PDB-SYS, SYMBA*, DOM-MULTI and SYMBB, which all solve similarly many problems (209–216 tasks). DOM-SINGLE, CARTESIAN and M&S fall further behind, but still solve more problems than HMAX, BLIND and LM-CUT. Looking at individual domains yields a similar ranking between the tested planners. As expected, exhausting the time limit is the main reason for failure for planners that perform costly calculations during the search, such as HMAX and LM-CUT. For the blind heuristic and single-abstraction heuristics (except for CARTESIAN), usually memory is the limiting factor. For the remaining approaches, the reasons for failure are mixed.

We now analyze our extensions of the abstraction heuristics. Their number of solved tasks is inversely proportional to the generality of the underlying abstraction class. This is a surprising result, as it seems reasonable to assume that the more general classes can represent more specific information when needed, resulting in more accurate heuristics. Indeed, CARTESIAN excels in terms of heuristic quality: its heuristic is often perfect when given enough time to refine the abstraction. Within our resource limits, however, this is only the case for the simpler problems and starting from a certain difficulty level in every domain, the resulting heuristics become less accurate, resulting in expanding many more states than the less general PDB and domain abstraction heuristics. The more general M&S heuristics, however, perform significantly worse in terms of expansions, both for simple and hard problems.

## Runtime Evaluation

One major advantage of the CEGAR-based approaches is their inherent mechanism to terminate once a plan is found within the CEGAR loop. Because of its capability to re-

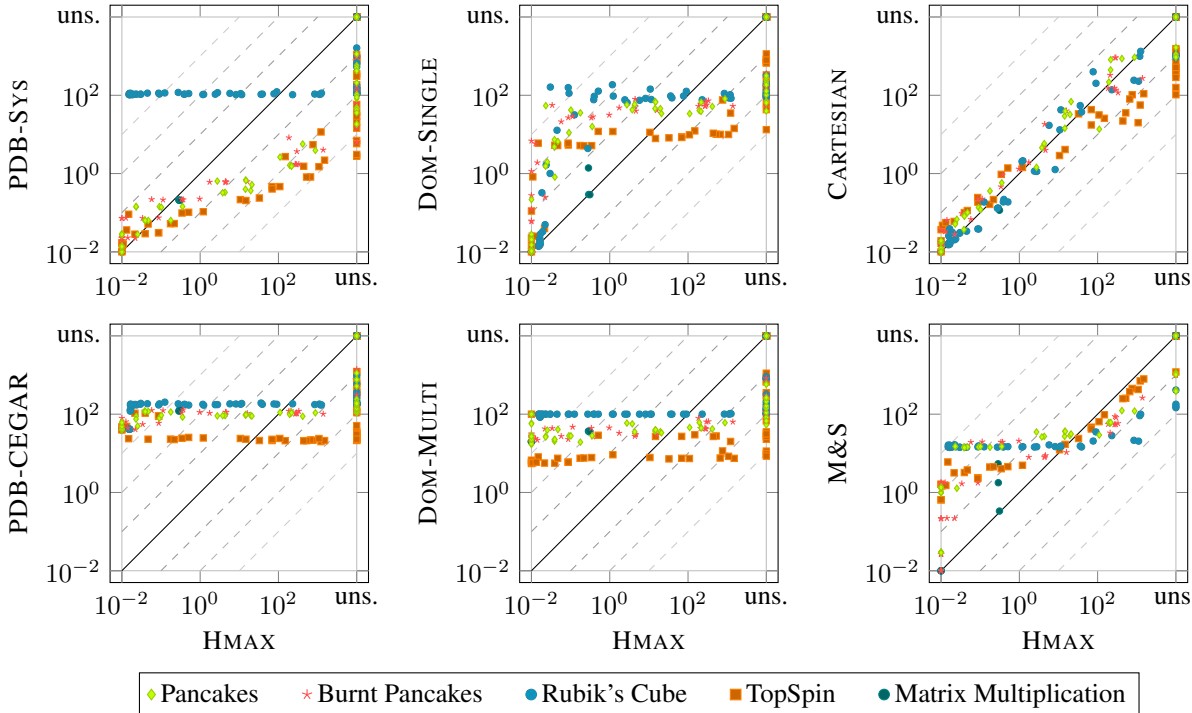

Figure 1: Runtimes in seconds split by domain: HMAX on the $x$-axis and our abstraction heuristics on the $y$-axis.

fine the abstraction very locally where needed, CARTESIAN benefits more from this compared to PDB-CEGAR, DOM-SINGLE and DOM-MULTI. In particular, the more restrictive abstraction classes generate larger abstractions faster. Since we limit their maximal abstraction size, they often terminate due to this limit before finding a solution.

Figure 1 compares the overall running times of our abstraction heuristics to HMAX. We use HMAX as the reference algorithm because its running time scales quite consistently with the difficulty level of the underlying planning task. We can see that there are many tasks where HMAX and CARTESIAN find solutions immediately. While DOM-SINGLE finds solutions quickly for some tasks, eventually it takes longer to compute the abstraction than it takes HMAX to find a solution. However, DOM-SINGLE can make up for this precomputation time with heuristic quality and has a somewhat constant solving time between 10 and 100 seconds, depending on the domain. This effect is more pronounced in the cases of DOM-MULTI and PDB-CEGAR, where precomputation times are clearly visible as horizontal lines for the different domains. The plots reveal that the task difficulty has limited influence on the search time of the planner once precomputations are completed. M&S also has a precomputation phase which only starts to pay off for problems where HMAX needs 10 or more seconds to solve them. Compared to the other methods, however, M&S requires more and more time as problems get harder.

In contrast, PDB-SYS consistently solves tasks faster than HMAX starting from 0.1 seconds. While PDB-SYS also requires precomputations, this does not show as clearly as for the other methods. This is most likely because the number of interesting patterns up to a certain size depends mostly on the number of variables in the problem, which gradually increases with the difficulty level for most of our benchmark domains. The only exception here is *Rubik's Cube* for which we use a constant-size cube with random walks of varying length to generate the tasks. When the number of variables becomes too large, precomputing all interesting patterns runs out of time or memory which is the reason why PDB-SYS does not outperform PDB-CEGAR coverage-wise, even though it has a clear speed advantage.

## Conclusions and Future Work

We extend the theory on common abstraction heuristics for classical planning. In particular, we study factored tasks, a generalization of SAS$^+$ featuring limited forms of disjunctive preconditions, conditional effects, and angelic nondeterminism. We compare implementations of our abstraction heuristics against other planning approaches supporting conditional effects on a newly created benchmark set. Our experiments reveal that PDBs are most successful in terms of overall coverage, while Cartesian CEGAR provides very accurate heuristics given enough resources for the precomputation. Since our benchmarks only feature the extension of conditional effects, it would be interesting to evaluate these heuristics on problems which also include disjunctive preconditions and angelic nondeterminism.

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
