# OpenReview forum: "Abstraction Heuristics for Factored Tasks"
_icaps-conference.org/ICAPS/2024/Conference — ICAPS 2024_

### Official Review · Reviewer_B7CK · 2024-01-14

**Significance And Importance:** 3
**Soundness:** 3
**Novelty:** 3
**Clarity:** 3
**Overall Evaluation:** 2
**Confidence:** 3

**Weaknesses:**

1: Minor weaknesses that are easily fixable.

**Contributions Of The Paper:**

The authors describe factored planning tasks, empirically study 5 interesting domains from the literature comprising such tasks,  and do so in the context of adapting a range of well known abstraction heuristics to the setting of factored planning tasks.

Adapted abstraction heuristics include: Pattern Databases, Domain Abstractions, Cartesian Abstractions, and Marge-and Shrink. Here, of course the detail of particular implementations (by the authors in the Scorpion planner, and in other tools for experimental comparison) of such abstractions is non-trivial, and the authors describe conceptual difficulties and how they choose to overcome those. Also, multiple abstraction settings for each broad class of abstraction are explored in the case of PDBs and Domain Abstractions. Of particular interest, and highlighted by the authors, is a new algorithm for finding Cartesian Abstractions.

**Ethical Considerations:**

(1) Not Applicable: The paper does not have any ethical considerations to address

**Nomination For Best Paper:**

No

**Questions For Authors:**

Question:-

On Line 250 authors report that DAs are generalisations of PDBs, with a short argument by construction given. But then on adapting DAs and PDBs two separate treatments are given, with PDBs apparently suffering so the authors require all the checks around Line 340. But it is noted that such checks are not required for DAs, as the issues addressed do not exist. It seems wonky, that the more general abstraction does not suffer issues of the less general abstraction. Could you please shed some more light on this?

[[post rebuttal]] -- many thanks for your answer to my above primary question. At the very least I think your "nutshell" comment should appear explicitly in your paper. With a wide readership overtime in mind, I also suggest your fuller explanation provided here would add value to the paper. Some details are already in there, so not a huge burden to integrate this explanation explicitly.

**Reproducibility:**

4: Authors promise to release code and domains (whichever apply).

**Strengths Of The Paper:**

The biggest contribution of this work is the implementation of the reported adaptations of heuristics to the factored tasks setting, and comparative empirical analysis performed. This is all new, relevant, and approached thoroughly. I was very impressed with the number of domains and problems experimented on (over 400 tasks, some of which are clearly quite challenging, and some of which do enable the authors to make some interesting conclusions regarding abstraction heuristics). The coverage and reported work related to comparing with symbolic and heuristic search algorithms is very good.  Runtimes contrasting each studied heuristic, by domain, against an HMAX baseline are also presented, along with some experimental conclusions from that comparative analysis. The provided information and analysis gives the reader some interesting information regarding bottlenecks and expected performance of evaluated techniques as they have been implemented. This is an good and acceptable experimental evaluation for our purposes, and this is a major aspect of the value of the work.

**Weaknesses Of The Paper:**

Weaknesses:-

In the front matter the authors highlight support for (angelic) nondeterminism and disjunctive preconditions. These seem like compelling features that are not empirically explored in any detail, and are left to future work.

I have been exposed to published data and arguments that suggest SAT-based planning is a compelling approach for makespan optimisation in settings such as these. That approach is missing from the experimental comparison. Some discussion on this topic, regarding any preliminary experimentation performed for example, is expected. [[post rebuttal]] not ideal this weakness is not to be addressed by authors, but in my evaluation acceptable to have this narrow scope for our purposes.

The details of the experimental environment are not given.

Some nitpicking issues include:

 - L385 and after, it is not clear from the language here, whether what remains an open question is about this work specifically, or more generally regarding CAs. On reflection, clearly this is a broader comment about the CA literature. That could be made more clear.

 - The authors talk about "real solution" (L471) without defining that.

 - The paper mixes discussion about costs, "shortest", and goal distance. Clearly this is just a goal distance paper.

 - There are grammatical issues.  L233 - "exploit that --> exploit ...the fact... that".

---

> ### Author Rebuttal · Authors · 2024-01-26
>
> Thank you for your review!
>
> Regarding your question: *semantically*, domain abstractions generalize projections. However, the relevant *syntactic* abstractions, as usually defined in other work, work differently: for projections, the idea is to project a variable "v" away by removing it entirely from the variable set, preconditions, effects, initial state and goals. For the special case of domain abstractions that correspond to projections, the corresponding operation is to change the domain of the variable to a singleton set {const} and replace all preconditions and effects of the form "v \mapsto x" with "v \mapsto const".
>
> This turns out to be equivalent in all cases *except* when we express an impossible condition on v. For a projection that discards "v", syntactic abstraction would replace a precondition like "v in \emptyset and w in {a, b}" by "w in {a, b}" because the precondition on "v" is simply removed, replacing an impossible precondition by a satisfiable one, which is a (wasteful) overapproximation. Viewed as a domain abstraction, we would replace every value of v by "const", but the precondition would then still be "v in \emptyset and w in {a, b}", so the impossible precondition gets preserved.
>
> In a nutshell: projections are a special case of domain abstractions semantically, but the usual definition of *syntactic projection* is not a special case of the usual definition of *syntactic domain abstraction*.
>
> In practice, it is of course possible to adjust the implementation of projection heuristics to deal with this case correctly. It is not of great practical relevance because trivial operators and trivially unsolvable tasks tend not to arise naturally, and if they do, they can be dealt with as simple special cases (discard trivial operators in a preprocessing step; report trivially unsolvable tasks as unsolvable without performing any search).

---

### Official Review · Reviewer_cXwv · 2024-01-18

**Significance And Importance:** 2
**Soundness:** 3
**Novelty:** 2
**Clarity:** 4
**Overall Evaluation:** 2
**Confidence:** 4

**Weaknesses:**

0: Minor weaknesses requiring some work to be addressed for the paper to be accepted.

**Contributions Of The Paper:**

This is a well-written paper that extends abstraction heuristics to more complex tasks containing a limited type of conditional effects. Other extensions are described but not tested, they are left as future work (disjunctive precondition and angelic nondeterminism).

Post-Rebuttal Comments:

Thanks for the rebuttal. After reading it and the other review and rebuttals, I have decided to raise my score to accept. The rebuttal makes a good case of why this paper's contributions, as they stand, are helpful for the community as a whole. I still think the technique is bit niche because it forces you to describe domains in SAS+ style. I would also have preferred the number of tested domains to be larger but I accept the authors' justifications.

In terms of small changes, I would ask the authors to make it clear in the claims that in order for this contribution to reach its potential, it requires an automated PDDL to factor in the SAS+ translator, which is future work. It would also be useful to add the CaveDiving domain to show another domain where the technique does not improve performance AND to explain why.

**Ethical Considerations:**

(1) Not Applicable: The paper does not have any ethical considerations to address

**Nomination For Best Paper:**

No

**Questions For Authors:**

1. Can you explain why your benchmarks are so focused on simple (in terms of modelling complexity) puzzle-like domains?
2. Have you considered how to modify the regular PDDL to SAS+ translator in FD to automatically generate tasks in your proposed extended SAS+ formalism? Please explain how this would be done.
3. If the translation is straight-forward, why did you not do it for this paper?
4. Can you provide examples of IPC domains where the extension would be useful?

**Reproducibility:**

4: Authors promise to release code and domains (whichever apply).

**Strengths Of The Paper:**

1. Well-written, easy to read and gives useful examples.
2. The paper extends several types of important abstraction heuristics: projections(PDBs), domain abstractions, and Cartesian CEGAR. An implementation for M&S is also provided using existing theory.  The PDB extension is quite straightforward, so the main contributions are extending domain abstractions and CEGAR.
3. It extends the very popular SAS+ formalism used by FD and derivatives.
4. Sufficiently Self-enclosed.
5. Related literature is well covered.

**Weaknesses Of The Paper:**

1. The advance does not cover all types of conditional effects, only those of factored tasks, i.e. the effects on variable x only depend on the value of x.
2. This is a planning paper, and hence, it would be better if it was domain-agnostic. All of the benchmarks but one are of the same "family" (permutation puzzles). The additional domain involves matrix multiplication, which is also a straight-forward permutation-type problem (for which all tested methods solve the exact number of problems; how is this a useful addition?). It is unclear how useful this extension is in practice to Planning in general, where most problems are not permutation-puzzle style.
3. No mechanism is provided to translate PDDL to the extended formalism. Domain authors are expected to write the problem in this enhanced SAS+ formalism, which is why the benchmarks are permutation-style problems which are easy to model in the modified SAS+ style using a scripting language like Python.
4. It is also unclear if an automated translation form PDDL for these domains would result in an SAS+ representation optimized to exploit the conditional effects enhancement.
5. The fact that maximizing PDBs does better than SCP reinforces the fact that the benchmark selection is biased towards a specific type of domain, which is not representative enough.
6. To summarize the previous points, it is a very niche improvement as it stands.

---

> ### Author Rebuttal · Authors · 2024-01-26
>
> Thank you for your review!
>
> Q1: We acknowledge the point, but there are also interesting differences:
> - Rubik's Cube: has a fixed number of actions each with the same number of effects.
> - Pancakes: the number of actions and effects depends on the problem.
> - TopSpin: there are always two actions, and the number of effects depends on problem size.
> - Matrix Multiplication: is not a permutation problem.
>
> The reason for the prevalence of permutation problems is that classical state-space search problems of this kind are primarily covered by the ICAPS and SoCS communities. For ICAPS, there is a chicken-and-egg problem: most domains are limited to STRIPS/SAS+ because this is what most algorithms support. In this paper, we try to to improve this situation. For SoCS, permutation problems seem to be the most popular benchmarks (apart from problems like MAPF that are best served by domain-specific algorithms).
>
> The overwhelming majority of ICAPS classical planning benchmarks are restricted to SAS+, so they are factored and hence within our scope, but already covered by the existing algorithms without requiring extension. There are a few exceptions that go beyond; for example, the CaveDiving domain is naturally factored but not SAS+. We briefly experimented with it and got similar results as for matrix multiplication: all planners solve the same problems. We are happy to include this domain in the paper.
>
>
> Q2+3: Thanks for the interesting idea! This is not straightforward, but has significant potential, as many common IPC domains could be expressed more compactly by making full use of factored representations (see next answer). But we really believe this would have to be a full paper of its own and could not simply be added to the existing paper within the given space limits.
>
>
> Q4: There are indeed domains in the IPC benchmarks which could benefit from being represented as factored tasks. For example, a logistics task with N trucks and N locations per city has O(N^3) drive actions in the grounded PDDL, STRIPS, and SAS+ formulations, but only O(N) as a factored task with our representation. This similarly applies to other transportation-style domains, but also to other domains like Schedule or Woodworking.

---

### Official Review · Reviewer_Kkee · 2024-01-22

**Significance And Importance:** 3
**Soundness:** 3
**Novelty:** 3
**Clarity:** 4
**Overall Evaluation:** 2
**Contributions Of The Paper:** 1. Generalization of abstraction heur…
**Confidence:** 3

**Weaknesses:**

1: Minor weaknesses that are easily fixable.

**Ethical Considerations:**

(5) Excellent: The paper comprehensively addresses all of the applicable ethical considerations

**Nomination For Best Paper:**

No

**Questions For Authors:**

1. Can you provide insights into the potential use of these heuristics in more intricate or unpredictable settings, like dynamic real-time systems?
2. How could these heuristics be incorporated into or interact with deep reinforcement learning frameworks?
3. Are there scenarios where the proposed heuristics might encounter scalability challenges, and how could these be addressed?
4. There is No comparison or discussion related to the PDDL representation, as this Arxiv paper compares the representation and performance of SAS+ and PDDL representations. https://arxiv.org/pdf/2307.13552.pdf
PDDL domain: https://github.com/ipc2023-classical/ipc2023-dataset/tree/main/opt/rubiks-cube
- The arxiv paper also talks about the performance of PDBs on problem solvability and compares across representations. What are they talking about in the context of PDBs and factored effect tasks (conditional effects in PDDL)? Are their results consistent with the ones mentioned in Arxiv's paper? Is there any reason for not citing this relevant work?

**Reproducibility:**

4: Authors promise to release code and domains (whichever apply).

**Strengths Of The Paper:**

1. Groundbreaking research in a primarily uncharted area that fills a significant void in the canon of planning literature.
2. The experimental evaluation provides a decent comparative study of different heuristic approaches.
3. Presentation clarity enables researchers and practitioners of AI planning to comprehend and apply complicated concepts easily on limited tasks.
4. To reduce the domain-specific knowledge needed for Pattern Databases (PDBs), the study introduces a generalization of abstraction heuristics for factored tasks.
5. This includes advances in algorithms such as CEGAR, which automate the refinement process and increase the applicability of heuristics in more fields.

**Weaknesses Of The Paper:**

1. Work has a narrowed focus to only factored tasks.
2. A section discussing applicability to other challenging and unpredictable planning domains is missing.
3. The methodologies explored and compared are limited. There has been a lot of work in the field of deep reinforcement learning, and a baseline comparison with some of those methods would have given a much clearer understanding of the work applicability and performance comparison

Examples of other Planning domains:
- Real-time strategy games, where the state of the game changes rapidly and unpredictably.
- Autonomous vehicle navigation with changing traffic patterns.
- Robotic systems

---

> ### Author Rebuttal · Authors · 2024-01-26
>
> Thank you for your review!
>
> Q1: Our focus is on classical planning, so we have not considered this so far. We think that using Cartesian abstractions might be the most suitable method among the ones we consider. Using Cartesian CEGAR for online abstraction refinement is probably a good choice because every single refinement step is very cheap. This has already been explored for SAS+ planning by Eifler et al. (AAAI 2019).
>
>
> Q2: Standard deep reinforcement learning approaches don't have access to a model, or learn a non-symbolic model from data. It is unclear how heuristics based on symbolic models could be used there. However, there is some work on feeding a symbolic planning model to a deep reinforcement learning agent (https://arxiv.org/abs/2203.00669).
>
>
> Q3: For the abstraction-based heuristics we consider there has been quite some work on making their computation fast and memory-efficient, but in general optimal classical planning remains a problem where no approach is able to solve the entire benchmark suite.
>
>
> Q4: In our experience, it is not so common in the ICAPS community to compare to non-peer-reviewed papers because the results in such papers have not been vetted by the research community. (By this, we don't mean to say that the cited paper is not valuable, and we hope that such a vetting will take place soon.) Also, the focus of our work is not to compare different representations of Rubik's Cube, but to investigate factored task models (which the Rubik's Cube model in the cited paper is not). For the purpose of comparing different Rubik's Cube domain models, it seems that the cited paper already provides this, and their work includes the model we present in this paper. (We cannot comment on this in more detail without compromising anonymity.)
>
> If we added the proposed comparison, this would cover the main contribution of the arxiv paper and thus make it much more difficult for the authors to eventually publish a peer-reviewed version of their work, as it would be hard for them to argue novelty.

---

### Meta-Review · Area_Chair_SQQt · 2024-02-05

**Recommendation:** Accept (Oral)
**Confidence:** 4

**Metareview:**

This paper extends abstraction heuristics to more complex tasks containing a limited type of conditional effects. Other extensions are described but are left as future work (disjunctive precondition and angelic nondeterminism). They describe factored planning tasks, empirically study 5 domains from the literature, in the context of adapting a range of well known abstraction heuristics to their setting. Adapted abstraction heuristics include: Pattern Databases(PDBs), Domain Abstractions(DA), Cartesian Abstractions(CA), and Merge-and-Shrink(M&S). The particular implementations (in the Scorpion planner, and other tools for experimental comparison) of such abstractions is non-trivial. They describe conceptual difficulties and how they are overcome. Also, multiple abstraction settings for each broad class of abstraction are explored in the case of PDBs and DAs. Of particular interest is a new algorithm for finding CAs.

Strengths
- Groundbreaking research in a mostly uncharted area that fills a void in planning literature.
- The experimental evaluation provides a comparative study of different heuristic approaches.
- Paper clarity enables readers to comprehend and apply complicated concepts easily.
- To reduce the domain-specific knowledge needed for PDBs, they introduce a generalization of abstraction heuristics for factored tasks.
- It includes advances in algorithms such as CEGAR, which automate the refinement process and increase the applicability of heuristics in more fields.
- It extends several types of important abstraction heuristics: PDBs, DAs, and Cartesian CEGAR. An implementation for M&S is also provided using existing theory. The PDB extension is straightforward, so the main contributions are extending DAs and CEGAR.
- It extends the very popular SAS+ formalism used by FD and derivatives.
- We were impressed with the number of domains and problems experimented on (over 400 tasks, some of which are quite challenging, and some of which enable them to make some interesting conclusions).
- The coverage of reported work relating to symbolic and heuristic search is very good.
- Runtimes contrasting each studied heuristic, by domain, against an HMAX baseline are also presented, along with some experimental conclusions from this analysis.
- The provided information and analysis gives the reader some interesting information regarding bottlenecks and expected performance.

Weaknesses
- It does not cover all types of conditional effects, only those of factored tasks, i.e. the effects on variable x only depend on the value of x.
- As a planning paper, it would be better if it was domain-agnostic. All of the benchmarks but one are permutation puzzles. The additional domain involves matrix multiplication, which is also a permutation-type problem (since all tested methods solve the exact number of problems; how is this a useful addition?). It is unclear how useful this extension is in practice to Planning in general, where most problems are not permutation-puzzle style.
- No mechanism is provided to translate PDDL to the extended formalism. Domain authors are expected to write the problem in this enhanced SAS+ formalism, which is why the benchmarks are permutation-style problems which are easy to model in the modified SAS+ style using a scripting language like Python.
- It is unclear if an automated translation from PDDL for these domains would result in an SAS+ representation optimized to exploit the conditional effects enhancement.
- The fact that maximizing PDBs does better than SCP reinforces the fact that the benchmark selection is biased towards a specific type of domain.
- In summary it is a niche improvement, because it forces you to describe domains in SAS+ style.

Suggestions to improve the final submission:
- We think it would be helpful if they added details to their paper regarding (i.e., their answer to a reviewer question): "projections are a special case of domain abstractions semantically, but the usual definition of syntactic projection is not a special case of the usual definition of syntactic domain abstraction."
- We would ask them to make it clear in the claims that in order for this contribution to reach its potential, it requires an automated extension to the SAS+ translator, which is future work. It would also be useful to add the CaveDiving domain to show another domain where the technique does not improve performance and to explain why.
- It would be good if the paper mentioned what they said in their rebuttal: "There are indeed domains in the IPC benchmarks which could benefit from being represented as factored tasks. For example, a logistics task with N trucks and N locations per city has O(N^3) drive actions in the grounded PDDL, STRIPS, and SAS+ formulations, but only O(N) as a factored task with our representation. This similarly applies to other transportation-style domains, but also to other domains like Schedule or Woodworking."

The paper does not need an ethical statement.

**Ethical Considerations:**

(1) Not Applicable: The paper does not have any ethical considerations to address